# Large-scale self-organization of reconfigurable topological defect networks in nematic liquid crystals

Yuji Sasaki[1], V.S.R. Jampani[2,†], Chiharu Tanaka[1], Nobutaka Sakurai[1], Shin Sakane[1], Khoa V. Le[2,†], Fumito Araoka[2] & Hiroshi Orihara[1]

Topological defects in nematic liquid crystals are ubiquitous. The defects are important in understanding the fundamental properties of the systems, as well as in practical applications, such as colloidal self-assembly, optical vortex generation and templates for molecular self-assembly. Usually, spatially and temporally stable defects require geometrical frustration imposed by surfaces; otherwise, the system relaxes because of the high cost of the elastic energy. So far, multiple defects are kept in bulk nematic liquid crystals by top-down lithographic techniques. In this work, we stabilize a large number of umbilical defects by doping with an ionic impurity. This method does not require pre-patterned surfaces. We demonstrate that molecular reorientation controlled by an AC voltage induces periodic density modulation of ions accumulated at an electrically insulating polymer interface, resulting in self-organization of a two-dimensional square array of umbilical defects that is reconfigurable and tunable.

[1] Division of Applied Physics, Faculty of Engineering, Hokkaido University, North 13 West 8, Kita-ku, Sapporo, Hokkaido 060-8628, Japan. [2] RIKEN Center for Emergent Matter Science (CEMS), 2-1 Hirosawa, Wako, Saitama 351-0198, Japan. † Present addresses: Physics & Materials Science Research Unit, 162a Avenue de la Faiencerie, University of Luxembourg, Luxembourg (V.S.R.J); Department of Chemistry, Faculty of Science, Tokyo University of Science, 1-3 Kagurazaka, Shinjuku-ku, Tokyo 162-8601, Japan (K.V.L.). Correspondence and requests for materials should be addressed to V.S.R.J. (email: venkata.jampani@uni.lu) or to F.A. (email: fumito.araoka@riken.jp) or to H.O. (email: orihara@eng.hokudai.ac.jp).

Control of the large-scale formation of functional micro and nano-patterns is attracting intense interest in the interdisciplinary field of materials science[1]. Self-organization in soft matter systems, such as colloids[2,3], block copolymers[4,5] and liquid crystals[6,7], is widely used for designing materials with emergent properties and for templating structures with new functionalities. Recently, fabrication of periodic patterns focuses on the integration of top-down (lithographic) and bottom-up methods because structures prepared solely by self-organization often lack sufficient long-range order, which is crucial for practical use.

Nematic liquid crystals (NLCs) are anisotropic fluids that already possess long-range orientational order of the long axes of constituent molecules, with the preferred direction called the director (**n**). An advantage of NLCs is that the director can be easily controlled by external electric fields owing to its dielectric anisotropy, as in current LC display applications. Generally, inhomogeneous director structures, including topological defects, have high elastic energy costs and they appear as uncontrollable and unstable features in the bulk[8,9] unless geometrically frustrated by a surface[10,11] as seen in droplets[12,13] and around colloids[14–16]. Although there are some experimental systems that exhibit spontaneous periodic patterns under electric and magnetic external fields[17–22], during heating[23] and in submicron thin films[24,25], these observations are mostly limited to one-dimensional stripes. Two-dimensional patterns[17,19,23,25,26] are relatively rare and are not formed over a large uniform area as reconfigurable, tunable patterns. Therefore, the stabilization of the complex director fields in NLCs is achieved by top-down lithographic approaches such as AFM scratch methods[27,28], nanoimprint lithography[29] and photo alignment[30–34] for multi-stable alignments[29,35], light diffraction gratings[30] and optical vortex applications[33,36–39]. However, the templates used in these lithographic techniques limit the reconfigurability and controllability of the system, and spontaneous self-organization is important for further exploitation of NLCs.

In this work, we demonstrate stable arrays of defects in NLCs without preparing a pre-patterned mask. Our approach is based on the spontaneous self-organization of NLCs through the standard reorientation of the director supported by an AC voltage, $V$. For homeotropically aligned NLCs with negative dielectric anisotropy, the director tends to orient perpendicular to the applied field above a threshold voltage, $V \geq V_{th}$. When the director tilt is induced toward the cell (horizontal) plane, degeneracy remains in the azimuthal angle, $\varphi(x,y)$, which results in the formation of a topological defect in the $xy$-plane, and $\varphi(x,y) = s\phi + c$ holds. Here, $s = \pm 1$ and $c$ is a constant value. These topological configurations are often referred to as umbilics[40,41]. Contrary to conventional coarsening behaviour[42,43], we show that doping with a small amount of an ionic compound leads to the formation and stabilization of a large number of umbilics in a square arrangement without annihilation. The size of the grid is tunable from several hundreds of micrometres to several micrometres, producing a high-density of defects. A large single domain is obtained spontaneously by the edge effect of the electrodes. Moreover, optical manipulation enlarges the uniform area to the millimetre scale. The arrangement of umbilics can be regarded as a soft two-dimensional crystal on the micrometre scale, which enables the direct observation of the moving dislocations in non-uniform arrangements of umbilics.

## Results

**State diagram of the micrographic appearance.** We begin by describing the observable textures. Sandwich cells, consisting of two parallel glass plates coated with indium-tin-oxide (ITO), are filled with NLCs (CCN-mn; *trans,trans*-4,4′-dialkyl-(1α,1′α-bicyclohexyl-4β-carbonitriles) (Fig. 1a). The ITO glass is spin-coated with an amorphous fluorinated polymer, CYTOP (CTX-809A, Asahi Glass Co.), dissolved in a fluorinated solvent (CT-Solv.180, Asahi Glass Co.) to induce the homeotropic anchoring[44,45]. An AC voltage of $V = V_0 \cos 2\pi ft$ is applied to the ITO electrodes in order to induce the reorientation of **n** (Fig. 1b,c). The micrographic appearance is studied as a function of frequency $f$ and amplitude $V_0$. Figure 2a shows four typical

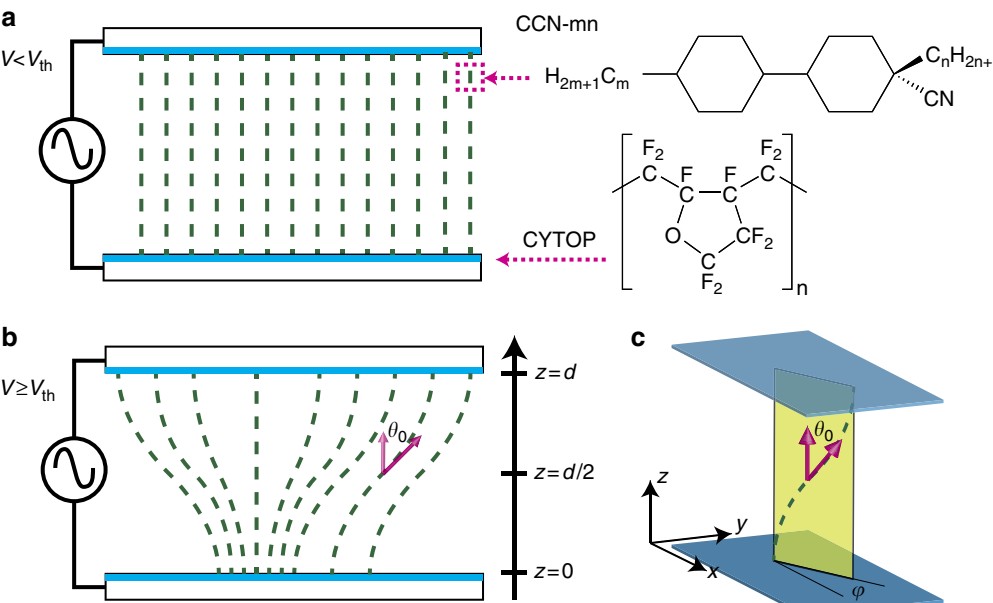

**Figure 1 | Schematic illustration of the director configuration in a sample cell.** (**a**) Cross-sectional view of the director configuration below the threshold voltage (homeotropic alignment). Chemical structures shown are the NLC (CCN-mn) and the alignment layer (CYTOP). (**b**) The director deformation above the threshold voltage. (**c**) The oblique view of the director field. It is to be noted that the director tilt $\theta_0$ is allowed for the arbitrary $\varphi$.

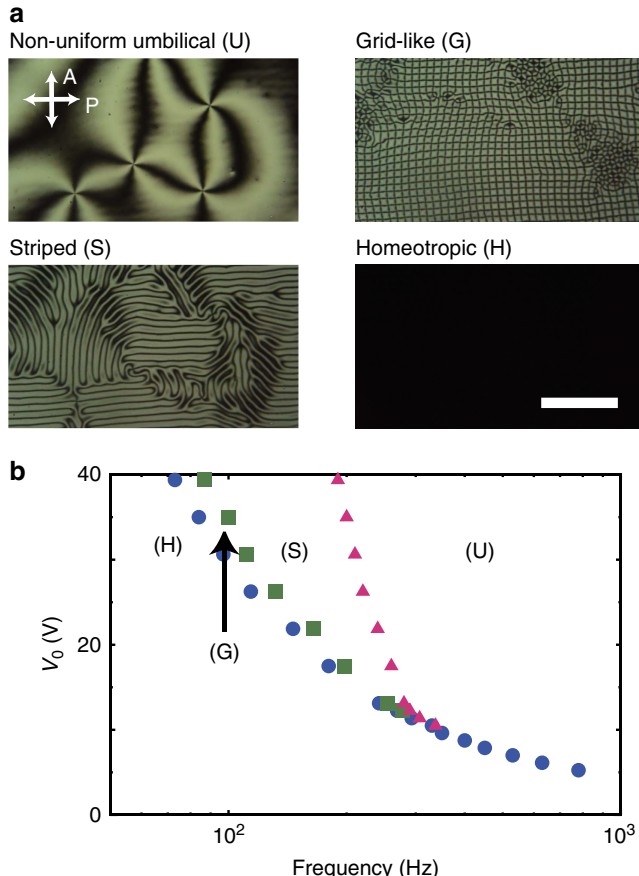

**Figure 2 | AC-voltage-dependent states.** (**a**) Four types of textures (umbilical (U), striped (S), grid-like (G) and homeotropic (H) textures) taken under crossed-polarizers. P and A denote the polarizer and analyser. All the images are taken with a constant $V_0 = 20$ V by varying the frequency. Scale bar, 100 μm. (**b**) The state diagram as a function of frequency and amplitude of AC voltages. The NLC used here is CCN-37 containing 1 wt% TBABE ions. The thickness of the NLC sample is 3.0 μm on average.

textures of an NLC, CCN-37, doped with an ionic compound, 1 wt% tetrabutyl ammonium benzoate (TBABE), taken under crossed polarizers. Figure 2b shows the state diagram plotted on the $f$-$V_0$ plane based on the observations of the 3.0-μm-thick cell. The details of the director configuration are addressed later (Fig. 4). At high-frequency, the well-known umbilical texture (U), consisting of randomly located ±1 umbilics, is observed. By decreasing the frequency gradually, a striped pattern (S) is formed from the U state. Further decrease of the frequency induces orthogonal strips, leading to a formation of a grid texture (G). Finally the dark texture of the homeotropic alignment (H) is observed in the low-frequency region. We stress that the emergence of these four states is qualitatively identical in the other cells with different cell thickness. As shown in Fig. 2b, although the present system is based on the standard Fréederickz transition, all the boundaries of the transitional voltages separating adjacent states increase sharply as the frequency decreases. Considering that the theoretical threshold voltage of $V_0 = \pi\sqrt{2K_3/\varepsilon_0|\Delta\varepsilon|}$ (ref. 46), where $K_3$ is the elastic constant of the bend deformation, is several volts using typical parameters[47], the observed $V_{th}$ curve in Fig. 2b is markedly different.

**Behaviour of threshold voltages**. We examine the behaviour of $V_{th}$, which denotes the transitional voltage between the

non-perturbed H alignment and the electric-field-induced G, S or U states, for several sample cells with various conditions. Typical results are summarized in Fig. 3a. Open symbols denote S and G states obtained by adjusting $V_0$ and $f$, whereas closed symbols show the U state above $V_{th}$. In our standard experiments, a solution containing 3 wt% CYTOP is used for spin-coating unless otherwise mentioned. The thickness of the alignment layer $l_s$ is ∼120 nm, as estimated by spectroscopic ellipsometry. In this condition, $V_{th}$ for the pure CCN-37 (used as received) is almost constant and small (magenta closed circles in Fig. 3a) due to the normal Fréedericksz transition. However, the ion-doped samples (open symbols in Fig. 3a) show a significant increase of $V_{th}$ at low frequencies at any cell thickness $d$, while a good agreement is seen for all the data at high-frequencies. These observations confirm that the sharp increase of $V_{th}$ is caused by the ionic contribution. To obtain more evidence, the thickness dependence of the CYTOP layer is examined, which leads to an important result explaining the $V_{th}$ behaviour on $l_s$. We dilute the CYTOP solution, for example, to 1.0 wt% (blue closed circles) so that the alignment layer becomes thinner ($l_s \sim 17$ nm) under the same spin-coating conditions, which also initially induce the uniform H state. In spite of the same ionic concentration and the same interface, $V_{th}$ substantially decreases compared with the data with open symbols and no further periodic G or S state pattern is formed. Moreover, the data for 0.8 wt% CYTOP solution ($l_s \sim 10$ nm, black closed circles in Fig. 3a) agree with those for the pure CCN-37 except for very low-frequency region. These decreases in $V_{th}$ indicate that a significant voltage drop occurs in the CYTOP layers due to the high resistivity which is larger than $10^{15} \Omega \cdot$ m (ref. 48) (http://www.agc.com/kagaku/shinsei/cytop/en/data.html), and the effective voltage on the NLC layer is essentially similar to all the cases. To support the finding, we also perform experiments using a surfactant monolayer of a silane-coupling agent, N,N-dimethyl-N-octadecyl-3-aminopropyltrimethoxysilyl chloride (DMOAP, Aldrich) and a polyimide layer (SE-1211, Nissan Chemical), which are well-known homeotropic surfaces. For SE-1211, effects of the spin-coating conditions are also examined, as described in the Experimental section. The behaviour of these homeotropic surfaces is similar to that of the pure NLC sample (magenta closed circles in Fig. 3a), which shows no substantial change in $V_{th}$. Furthermore, neither the G nor S textures appear. These results also prove that the increase of $V_{th}$ is responsible for the pattern formation caused by electrical insulation, that is, the field-screening effect of the ionic localization in the vicinity of the CYTOP layer (Fig. 3b). Thus, the underlying physical origin of $V_{th}$ curves is the Fréedericksz transition even though the variation of $V_{th}$ depends on the thickness of the CYTOP layer, $l_s$. In other words, the bulk reorientation by the dielectric property is essential in this phenomenon. We note that polarization microscopy also excludes the possibility of the director reorientation at surfaces due to the surface polarization effect, that is, there is not observed any typical cloudy pattern that depends on the surface polarity[49,50]. The possibility of the flexoelectric instability is also excluded for the G or S textures. Such flexoelectric domains can only be observed in NLCs with low dielectric anisotropies and the applied electric field must be DC or AC at low frequencies[51]. Besides, the contribution of electrohydrodynamics can be ruled out because the tracer particles immersed do not exhibit a motion due to the LC flow. In fact, the electrohydrodynamic convection (EHC) pattern emerges, overlapping with the G texture for a proper cell thickness, as described later (Fig. 8b). Moreover, comparing the data denoted by open symbols in Fig. 3a shows that the $V_{th}$ curve shifts to the low-frequency side as cell thickness $d$ increases. The tendency is also distinct from that caused by the polar surface

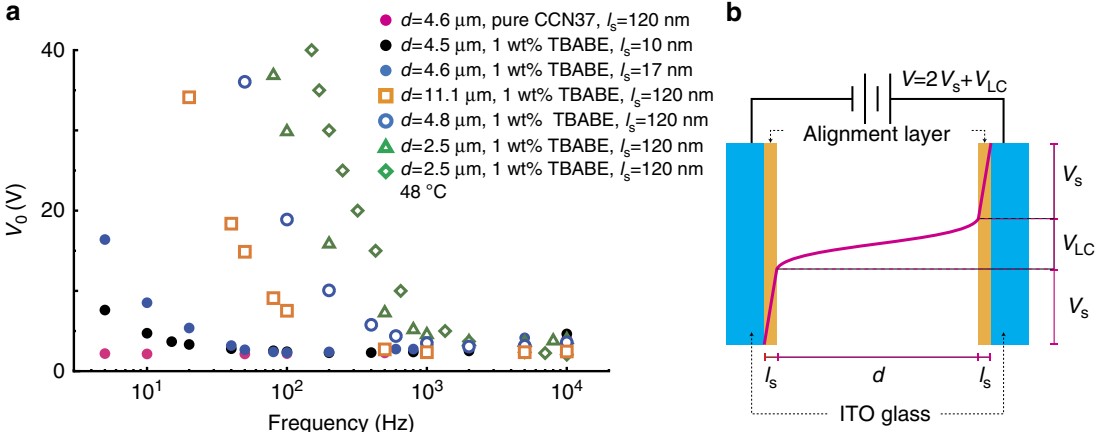

**Figure 3 | Behaviour of threshold voltage.** (**a**) The threshold voltages for the director reorientation from the homeotropic alignment. The data expressed with open symbols exhibit the grid-like (G) and striped (S) states at lower frequency. On the other hand, the data for the closed circles show only the non-uniform umbilical texture (U). The spin-coating condition is the same for all the data. 3 wt% CYTOP solution is used for spin-coating in all experiments except for the data denoted by closed blue and black circles respectively in which 1.0 wt% and 0.8 wt% CYTOP solutions are used to make a thinner alignment layer. (**b**) A schematic illustration to explain the voltage drop in the sample cell. The thicknesses of NLC and CYTOP are denoted with $d$ and $l_s$, respectively. The solid magenta curve denotes that a substantial voltage drop occurs not only in the NLC layer ($V_{LC}$) but also in the CYTOP alignment layer ($V_s$) of the sample cell exhibiting (G) and (S) states.

instability[49,50]. Furthermore, as the temperature increases, the G state returns to the H state, namely the $V_{th}$ curve shifts to the high-frequency side. These results are also consistent if the ionic dynamics is considered and a more detailed description of the role of CYTOP is provided in the Discussion.

**Spontaneous formation of a large single domain**. The G state is usually accompanied by multi-domains (Fig. 2a). Therefore, in addition to elucidating the director field, realizing a large single domain is of particular interest because it has huge potential applications. We present a method for creating single domains by combining self-organization with a simple top-down approach. Two glass substrates with stripe-patterned ITO electrodes typically several hundreds of microns wide are prepared and they are placed so that the ITO stripes cross (Fig. 4a). Then, we apply an AC voltage to the square regions of the overlap of the ITO stripes. In this experiment, $f$ is gradually increased with a constant $V_0$ to transform the H state to the G state. First, uniform arrangements of umbilics are spontaneously formed along the edges of the square region (Fig. 4b, Supplementary Movie 1). These initial umbilics trigger the epitaxial growth of a unidirectional single domain of the umbilics, and the G domain spreads over the whole square region of the overlapped electrodes. About 1,000 defects are packed with a regular spacing in Fig. 4c. To fill the area completely requires several tens to a few hundred seconds, depending on the quality of the sample cells. In our experimental conditions, this approach can be used up to the submillimeter scale.

The micrographic appearance is changed dramatically by rotating the crossed polarizers (Fig. 4c–e). When the direction of the extended lines connecting the neighbouring umbilics is parallel to the polarizers, a square lattice pattern is observed. A slight rotation of the polarizers (Fig. 4e) allows the topological nature of umbilics to be observed. The rotational directions of the four brushes of the adjacent umbilics are opposite, reflecting the $\pm 1$ states. The contrast of the image is inverted by rotating the polarizers by 45°. Figure 4f,g shows the effects of inserting a full-wave retardation plate ($\lambda$-plate) into the setup in Fig. 4c,d, respectively. The blue (added retardation) and bright magenta (subtracted retardation) regions imply that the director field

around $+1$ umbilics is considered to be a radial type rather than spiral one[40]. To support this consideration, fluorescent confocal polarizing microscopy observations are performed (Supplementary Fig. 1). This technique enables the director mapping by the intensity distribution of the polarized fluorescence from a doped emitter molecule transmitted through a polarizer inserted in the optical path[52,53]. An image taken at the middle plane of the cell agrees with our microscope observations. These observations lead to a texture consisting of two types of umbilics arranged in squares (Fig. 4h). In the present type of electrodes, the corner always has a hyperbolic hedgehog defect with $-1$ strength because of the topological constraint of the director field (Supplementary Fig. 2). The director deformation in the z-direction means that lens effects can be observed qualitatively by moving the objective focal plane of microscope (Supplementary Fig. 3).

**Direct manipulation of defect arrays by laser irradiation**. We show another important method for actively obtaining a large single domain without the edge effect of electrodes. We use an optical tweezers technique with an Nd-YAG laser (1,064 nm) to manipulate the local structure[54]. Owing to light-induced heating, the director in the laser spot returns to the perpendicular alignment, which can be seen as a dark spot (Fig. 5b and Supplementary Movie 2). After removing the laser spot, the director finds a more stable configuration while recovering its tilt angle. Thus, scanning the laser spot allows umbilics to rearrange to form uniform arrays (Fig. 5c). We can eventually create a domain on the sub-square centimetre scale (Fig. 5d). A larger scale would be possible by preparing ideal cells under clean room conditions.

Laser tweezing can be used to create or erase umbilics at arbitrary positions in the S state, which appears on the high-frequency side of the G state (Fig. 2). We use square electrodes, the edges of which form a regularly ordered S state (Fig. 6a). After preparing a single G domain, we adjust $f$ at a fixed $V_0$ to the conditions where the S state is slightly preferred over the G state. Then, the umbilics are irradiated with a laser to induce homeotropic alignment. After switching off the laser, the defects no longer appear and instead the more stable S state is obtained

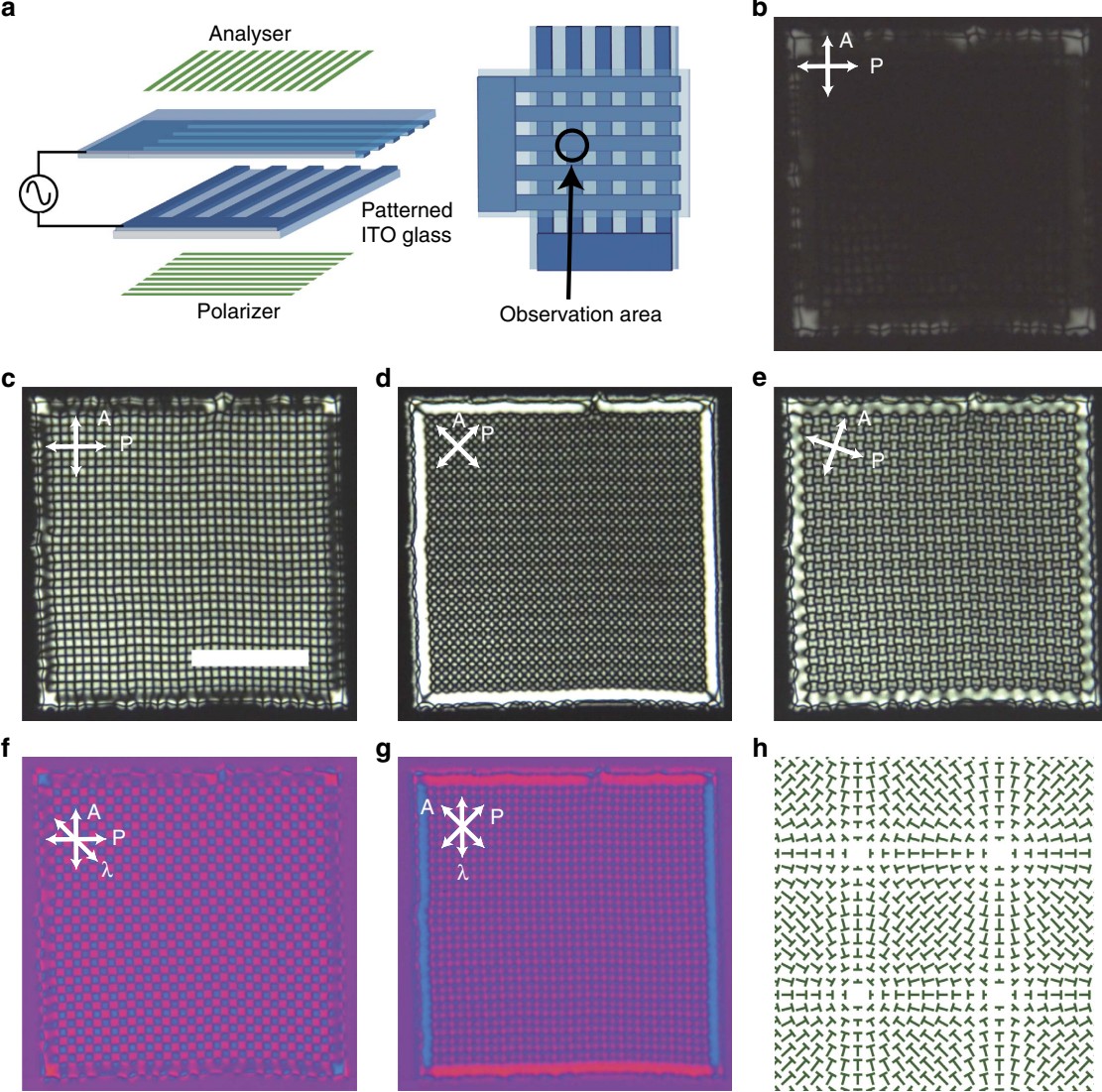

**Figure 4 | Template-assisted self-organization of square arrays of defects.** (**a**) Schematic illustration of sample cell with patterned ITO electrodes for inducing a single domain of the grid-like texture. (**b**) The initial stage of the formation of the grid-like texture from the edge of electrode. The frequency of the electric field is gradually increased. The contrast and brightness of the image (**b**) is different from (**c–e**) for visibility. (**c–e**) A single domain of defects array spontaneously obtained in an epitaxial way. These images shown are taken under different crossed-polarizers. (**f,g**) Textures with the insertion of a full-wave plate, denoted by λ. The NLC used is CCN-37 and the experimental condition here is $V_0 = 17.5$ V, $f = 110$ Hz. The cell thickness is 4.9 µm. Scale bar, 200 µm. (**h**) The schematic illustration of the director profile in defects array.

(Supplementary Movie 3). Once the S state is prepared, we can create artificial isolated umbilic arrays surrounded by the S domain (Fig. 6b). The whole area can be transferred from the S state to the G state (Supplementary Movie 4). Importantly, the S state can be obtained again by erasing umbilics, and thus the process is fully repeatable.

**Flow-induced striped texture.** Mechanical flow also affects the director field, **n**, substantially. When the NLC sample is introduced into the cell, the effect of capillary flow on the texture is visible. Here, we present an observation for a typical cell thickness of 4 µm. The flow speed cannot be controlled in our experimental conditions, because it depends on the cell gap and the observation area. Poiseuille flow occurs and the director profile can be regarded as two halves of the shear flow region[55]. The G structure changes to the S state as the sample flows during the injection (Supplementary Movie 5). The flow speed is

estimated to be $\sim 8$ µm s$^{-1}$ from observation of the moving umbilics. The action of the flow aligns **n** along the flow direction and simultaneously distorts it in the orthogonal direction. This can be explained by investigating the effect of Poiseuille flow on the umbilics[56]. To apply simple shear, we prepare a setup consisting of a cell with its upper substrate fixed to a motorized translation stage without using spacers. The series of snapshots in Fig. 7 (Supplementary Movie 6) shows that the shear flow strongly affects the texture. Even this simple experiment demonstrates interesting properties. The lines of the grids normal to the shear direction degenerate and disappear, whereas those parallel to the shear directions remain (Fig. 7a). Once the shear is stopped, the G state is spontaneously recovered (Fig. 7b). Comparing the two adjacent lines normal to the shear direction shows that their behaviour is different. The position of the defects on the white dashed lines in Fig. 7a,c is almost unchanged under shear flow and the motion of the adjacent grid lines is opposite depending on the flow directions. This is because the bottom

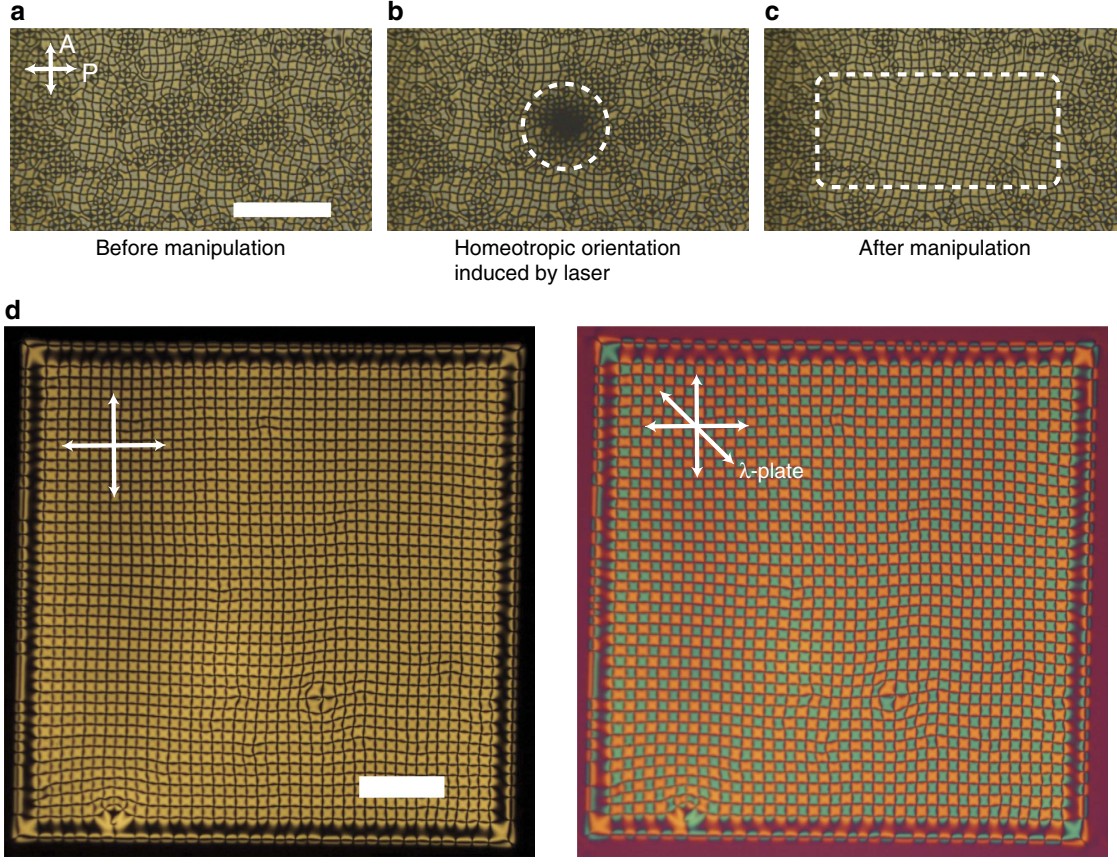

**Figure 5 | Creation of a single domain by an optical manipulation.** (**a**) A multi-domain structure of grid-like pattern which can be obtained by switching on an electric field suddenly. (**b**) The homeotropic alignment induced by the irradiation of laser light marked with a white dashed circle. (**c**) A uniform domain created by the optical manipulation. The cell thickness is 3.0 μm. (**d**) A large single domain of grid-like texture obtained by the help of optical manipulation. The cell gap is 5.9 μm. $V_0 = 32$ V. The NLCs used here are the 1:1 mixture of CCN-47 and CCN-55. Scale bars, 200 μm.

substrate is fixed (Fig. 7d). The blue and yellow strips exchange positions depending on the shear direction (compare the bottom figures of Fig. 7a,c). These behaviours are qualitatively consistent with the properties of umbilics. Details of the effect of mechanical flow on the G state will be reported in future work.

**Tunable grid spacing**. The spacing between umbilics can also be controlled. The grid size is measured by varying the cell thickness, $d$. Figure 8a shows that the size is almost proportional to $d$, which is tunable from several to hundreds of micrometres (Supplementary Fig. 4). This means that a thinner cell can generate high-density umbilics (Supplementary Fig. 5). For thick cells (for example, thicker than 20 μm in our experiments), the change in birefringence is clearly observed by polarization microscopy as the frequency increases (Supplementary Fig. 6). This is because the net birefringence is increased by the increase of the tilt angle, $\theta_0$, in the bulk (Fig. 1b). Further increases of the frequency cause electrohydrodynamic convection (EHC) (Fig. 8b). This is reasonable because EHC occurs for planar and homeotropic anchoring[57,58]. Even in the EHC region, the grid can be maintained with good stability. Because the director orientation is normal to the hydrodynamic rolls, the topological strength of umbilics is easily visualized by the direction of the rolls. This additional periodic modulation of the birefringence would offer interesting optical properties.

The size of the grid also depends on $V_0$; the spacing increases with $V_0$. Of course, $f$ must be decreased simultaneously in accordance with Fig. 2. This property can be used to control of

the number of grids in a single area dynamically (Fig. 8c and Supplementary Movie 7). The edge effect of the electrodes is important and helps to maintain the structure. Particularly in narrow electrodes, the high elastic energy cost due to the non-uniform deformation quickly relaxes to the single domain. Thus, the edges help produce a reproducible pattern when the field is switched on and off. We also confirm that even numbers of squares (or odd numbers of umbilics) are allowed inside the area because of the unique director configuration at the corner (Supplementary Fig. 2).

The variable grid spacing provides an interesting feature even for inhomogeneous arrangements. So far, we are mainly focusing on obtaining uniform square arrangements of umbilics. However, imperfect arrangements of umbilics often form dislocations like atomic crystals. This is a unique feature that is not observed in other analogue systems, such as two-dimensional bubble rafts[59], because in our system liquid crystal umbilics mimic the atoms in crystals. Dislocations at a grain boundary can be generated artificially by using the acute corner of an oblique cross of the ITO strips (Fig. 9a). The umbilics tend to align along the edge of the electrodes and they prefer to be packed in squares; thus, grain boundaries appear. Three different cross-polarized conditions show three domains: two domains near the electrode edges and a central domain. Although the present system adds neither tensional nor compressional stress to the defect array in the cell (horizontal) plane, similar to the bubble raft system, frustration can be induced in the system by changing the grid size. Figure 9b shows a visualization of moving dislocations that are trying to reduce the frustration. The electric field is suddenly

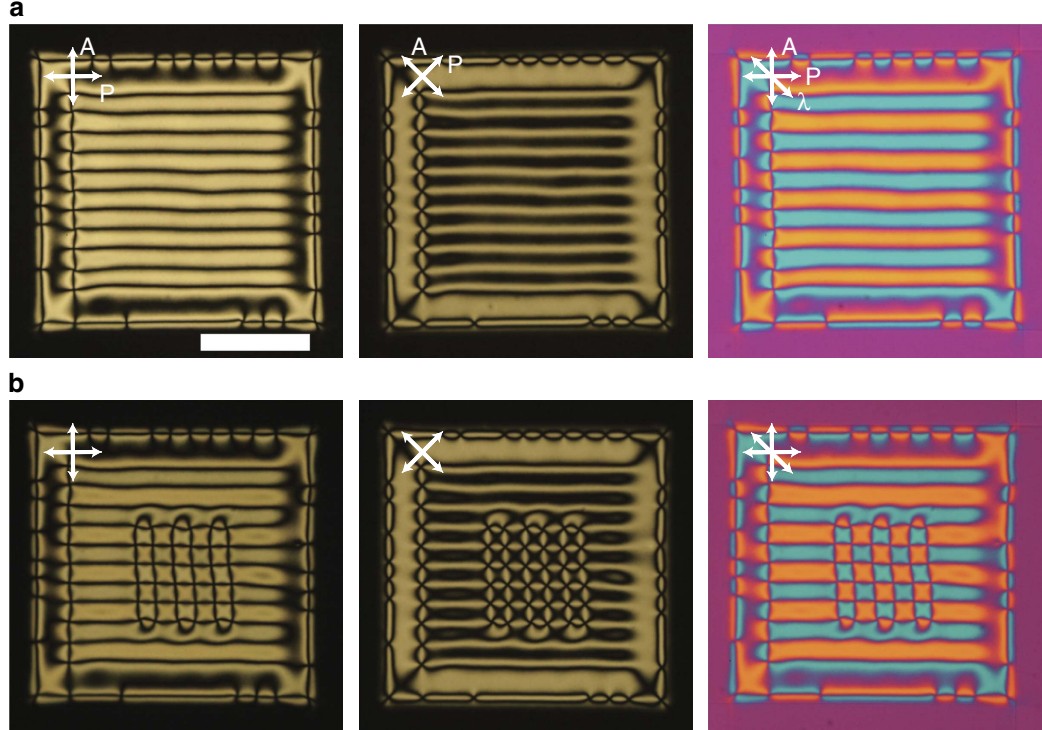

**Figure 6 | Creation of a grid-like texture on a striped pattern.** (**a**) Typical microscopic appearance of a single domain filled with striped pattern under different polarization conditions. (**b**) An isolated domain of grid-like pattern created on the stripe domain with an optical manipulation. CCN-37 is used and the average cell thickness is 3.2 μm. Scale bar, 100 μm.

changed to a higher $V_0$ and a lower $f$ to increase the grid size (Fig. 8c), and then a lower density of umbilics is required to reach a stable state. Thus, because of the imposed elastic frustration, a pair of umbilics annihilates to generate two dislocations, and then these dislocations move apart to relax the structure. The time course is shown as snapshots in Fig. 9b (Supplementary Movie 8). The dislocation moves in the normal direction of the glide plane (or line). This is natural considering the energetics because this motion reduces the density of umbilics. This is a distinct difference from the atomic crystals where the dislocation moves parallel to the direction of the glide line because the atoms never annihilate.

## Discussion

The G and S states can be obtained by the combination of dopant ions and a CYTOP layer with a proper thickness. Here, in this study, we consider the field-screening effect as mentioned above. In order to explain the sharp increase of $V_{th}$ at low frequencies, we use the configuration shown in Fig. 3b to calculate the voltage applied to the cell. The dielectric constant is $\varepsilon_{LC}$, the electric conductivity is $\sigma_{LC}$, and the thickness of the NLC is $d$, and the corresponding terms for the alignment layer are $\varepsilon_s$, $\sigma_s$, and $l_s$, respectively. The anisotropy of $\varepsilon_{LC}$ and $\sigma_{LC}$ is not considered and $\sigma_{LC}$ includes the ionic contribution. The applied voltage $V = \mathrm{Re}\left[\tilde{V}e^{i\omega t}\right]$ is written as $V = V_s + V_{LC}$, where $V_s = \mathrm{Re}\left[\tilde{V}_s e^{i\omega t}\right]$ and $V_{LC} = \mathrm{Re}\left[\tilde{V}_{LC} e^{i\omega t}\right]$ are the voltage drop in the alignment layer and in the NLC layer. Using the complex conductivity, $\tilde{\sigma}_{LC} = \sigma_{LC} + i\omega\varepsilon_{LC}$ and $\tilde{\sigma}_s = \sigma_s + i\omega\varepsilon_s$, a relation $\tilde{\sigma}_{LC}\tilde{V}_{LC}/d = \tilde{\sigma}_s\tilde{V}/2l_s$ should hold due to the conservation of current density. The complex amplitude of the threshold voltage applied to the cell is described as $\tilde{V} = \{2(l_s/d)(\tilde{\sigma}_{LC}/\tilde{\sigma}_s) + 1\}\tilde{V}_{LC}$. Low and high-frequency limits are evaluated by $\tilde{\sigma}_{LC}/\tilde{\sigma}_s = (\sigma_{LC} + i\omega\varepsilon_{LC})/(\sigma_s + i\omega\varepsilon_s)$. Here CYTOP has a high

volume resistivity of $1/\sigma_s > 10^{15}\,\Omega\cdot\mathrm{m}$ and a small dielectric constant of $\varepsilon_s = 2.0\text{–}2.1\varepsilon_0$ (ref. 48). The resistivity is at least around an order of magnitude greater than that of the polyimide surface ($10^{10}\text{–}10^{13}\,\Omega\cdot\mathrm{m}$) (ref. 60), while the dielectric constant is still comparable to that of polyimide and LCs, that is, $\varepsilon_s \sim \varepsilon_{LC}$. It is a reasonable assumption that the electrical potential, $V_{LC}$, necessary for the reorientation of **n** is almost the same as the Fréedericksz transition voltage of the pure NLCs shown in Fig. 3a, namely the rest of the applied voltage, $V - V_{LC}$ is used for the voltage drop in the alignment layer. We also note that $l_s/d \ll 1$ holds because the thickness is $l_s \sim 120\,\mathrm{nm}$ in our standard spin-coating condition. Then the high-frequency region gives $V \sim \{2(\varepsilon_{LC}/\varepsilon_s)(l_s/d) + 1\}V_{LC} \sim V_{LC}$, because the dielectric constant is dominant. However, the conductivity becomes important at the low-frequency side and $V$ can be approximated as $V \sim \{2(\sigma_{LC}/\sigma_s)(l_s/d) + 1\}V_{LC}$, which is greatly affected by the amount of doped ions $\sigma_{LC}$. Our preliminary evaluation of the effective conductivity of CCN-37 is of the order of $10^{-6}\,\Omega^{-1}\,\mathrm{m}^{-1}$ at 1 kHz, which is measured without using an alignment layer. Thus, the main reason can be assigned to the increased value of $\sigma_{LC}$. This supports our experimental data that show thicker cells lower the threshold voltage at the same frequency. Eventually the surface charge density, $\rho_s$, given by $\rho_s = \varepsilon_0\{\varepsilon_{LC}(\sigma_{LC}/\sigma_s) - \varepsilon_s\}(V_s/l_s)$ in the low-frequency limit, becomes high. We can speculate that a high surface charge density is achieved on CYTOP after doping with ionic materials and that it creates a spatial distribution near the interface. The expression of $\rho_s$ also supports the temperature dependence because the ionic localization around the surface develops faster owing to the increased mobility, $\mu \propto \sigma_{LC} = \sigma_0\exp(-W/k_B T)$ (ref. 46).

Since the qualitative behaviour of $V_{th}$ is explained by considering the CYTOP surface, here, we present a detailed model that can reproduce the transition to the G state from

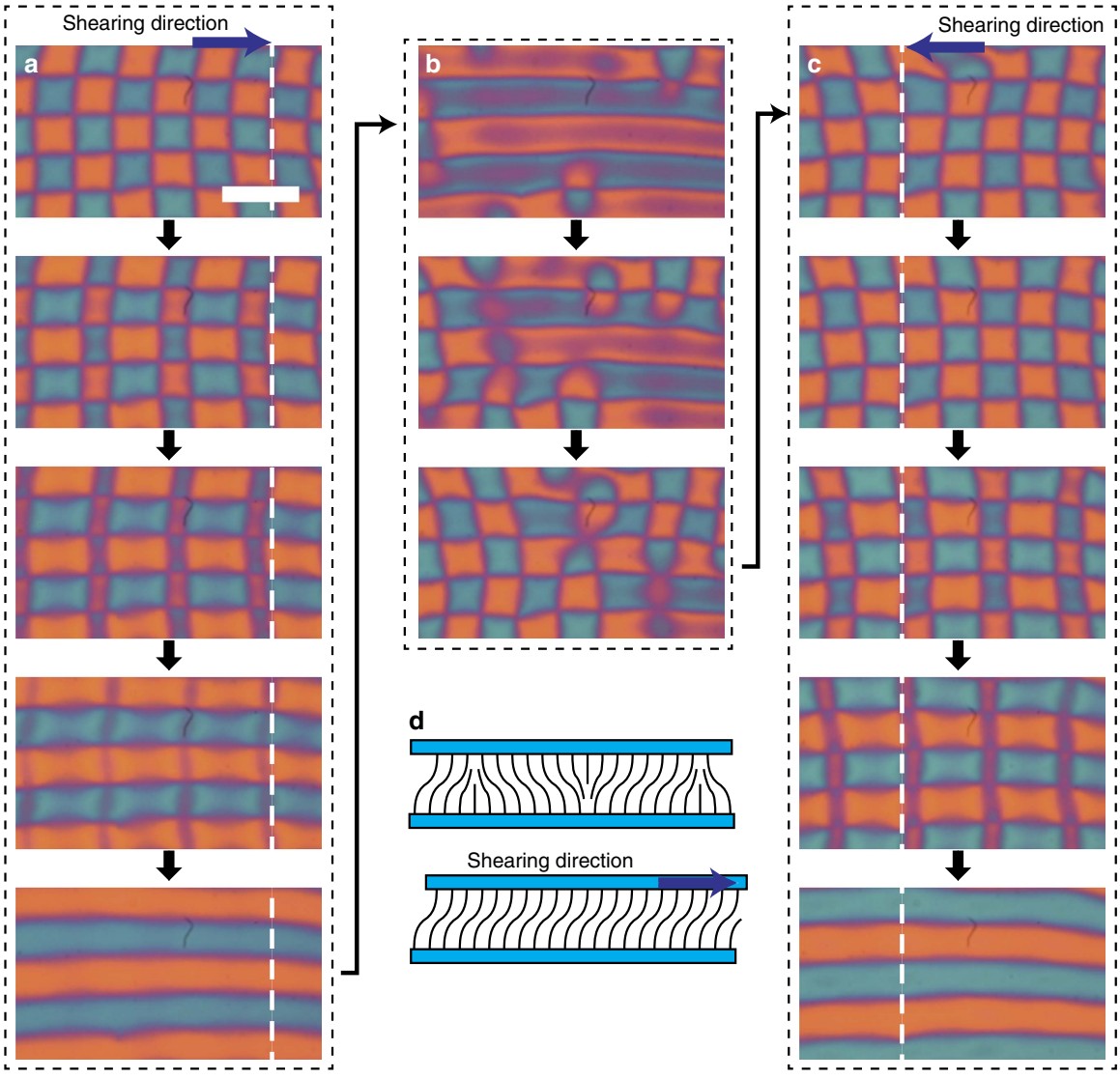

**Figure 7 | Effect of a shear flow on the grid-like texture.** The snapshots for the effect of the shear flow for the grid-like texture. (**a,c**) The texture under shear flow. Dashed lines are the guide to the eye. (**b**) Recovering the grid-like texture. (**d**) Shematic illustration for the cross section of the director field. The horizontal allows show the direction of shear flow. The NLC used is 1:1 mixture of CCN-47 and CCN-55. Scale bar, 100 μm.

the H state prior to achieving the U state. Based on the observations, the effects of the doped ions (TBABE) and the insulating layer (CYTOP) are considered. The details are provided in the Method section. We set the origin of the $z$-axis ($z = 0$) at the center of the nematic slab (Fig. 10a). The NLC-CYTOP interfaces and the CYTOP-electrode interfaces are positioned at $z = \pm d/2$ and $z = \pm h/2$, respectively. We focus on calculating the threshold voltage from the H to G states. In our system, the electric potential is given as $\phi = 0$ at $z = -h/2$ and $\phi = V_0 \cos\omega t$ at $z = h/2$, with $\phi = \mathrm{Re}[\tilde{\phi}e^{i\omega t}]$. The slightly tilted **n** is expressed as $\mathbf{n} = (\delta n_x, \delta n_y, 1)$ and the corresponding potential becomes $\tilde{\phi} = \tilde{\phi}_0 + \delta\tilde{\phi}$. The translational symmetry in the $x$-$y$ plane allows us to write the solution for the grid pattern as

$$\delta n_x(x, y, z) = \theta(z)\cos qx \sin qy, \quad (1)$$

$$\delta n_y(x, y, z) = \theta(z)\sin qx \cos qy, \quad (2)$$

$$\delta\tilde{\phi}(x, y, z) = \tilde{\psi}(z)\sin qx \sin qy, \quad (3)$$

where the grid pattern described by equations (1)–(3) should be rotated by 45° when it is compared with the photos shown in

Fig. 4. The threshold voltage can be regarded as a function of $q$, that is, $V_0(q)$. The real threshold voltage, $V_{th}$, is given by the minimum value of $V_0$. A local minimum at $q = 0$ corresponds to the transition to the U state, namely, the normal Fréedericksz transition, and another minimum at $q \neq 0$ corresponds to the transition to the G state. Because $V_{th}$ depends on the frequency of the applied voltage, permittivities, conductivities, elastic constants and thicknesses of NLC and insulating films, the numerical calculations based on the continuum theory of NLCs are performed by using the typical material constants for the NLC and CYTOP, and our experimental conditions for $d$ and $l$. The conductivities $\sigma_\parallel$ and $\sigma_\perp$, which depend on the concentration of dopant ions, are chosen to reproduce the frequency dependence of $V_{th}$ in Fig. 2b. The behaviour of $V_0(q)$ at different typical frequencies is shown in Fig. 10b. At low frequencies of $f = 70$ and 200 Hz, the G state is more stable than the U state since the minimum of $V_0$ is at $q \neq 0$. However, the two local minima of $V_0$ at $q = 0$ and $q \neq 0$ become close at 200 Hz. As the frequency is increased, they become equal (457.5 Hz) and eventually the U state becomes more stable (550 Hz). Calculating $V_0(q)$ by varying the frequencies allows us to plot $V_{th}$ as a function of $f$ (Fig. 10c).

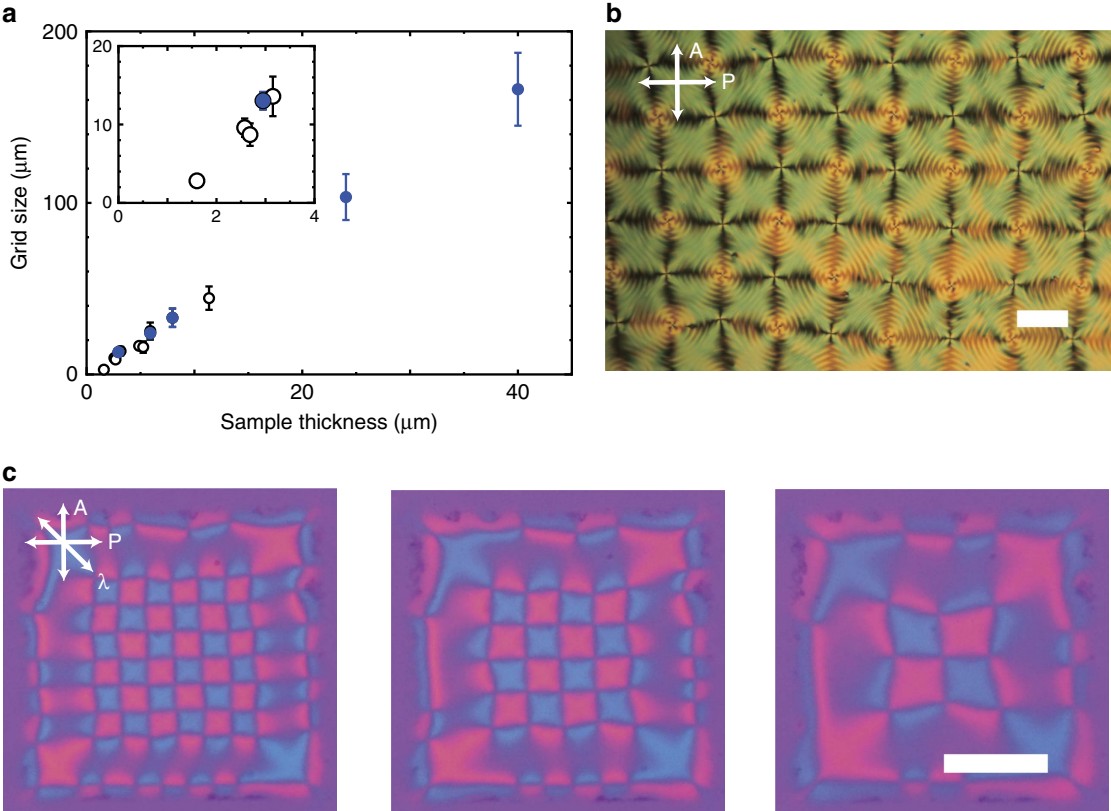

**Figure 8 | Tunable grid spacing by sample thickness and applied voltage.** (**a**) Distance of adjacent defects depending on the sample thickness $d$ and electric field strength $V_O$. The vertical bars are obtained when $V_O$ is changed (from $V_O \sim 10\,V$ to 40 V). Larger array size in the bar corresponds to higher $V_O$ and the frequency is adjusted in each case. The plotted data are the mean value of the minimum and maximum grid size. Closed circles are the data for 1:1 mixture of CCN-47 and CCN-55, and open symbols are for CCN-37. Because of experimental limitations, the range of the thickness used is above 1 μm. We note that there is no significant difference in the grid size between these two NLC samples. (**b**) A texture accompanying an electro-hydrodynamic convection observed in a relatively thick cell with $d = 24\,\mu m$. The mixture of CCN-47 and CCN-55 is used. (**c**) Controllable numbers of arrays in a narrow area. The left, middle, and right micrographs show arrays of $6 \times 6$ ($V_O = 17.5\,V$), $4 \times 4$ ($V_O = 26.2\,V$), $2 \times 2$ ($V_O = 39.3\,V$), respectively. Here CCN-37 is used and the average cell thickness is 3.7 μm. Scale bars, 100 μm.

The magenta lines are the boundaries of the H–G transition with $q \neq 0$ and the black lines are those of the H–U transition with $q = 0$. The blue solid circles indicate the intersection of the two lines. We cannot reproduce the G–S and S–U transitions because our numerical results are limited to the linear stability analysis. (i) in Fig. 10c corresponds to Fig. 2, where we use the same material constant values as in Fig. 10b. Good agreement is obtained, though the numerically obtained position of the boundary between the H–G and H–U transitions is located at a little bit higher frequency compared with the experiment. When the conductivities $\sigma_{\parallel}$ and $\sigma_{\perp}$ are decreased ((ii) in Fig. 10c), the boundary between the H–G and H–U transitions shifts to a lower frequency, implying that the grid pattern is destabilized by the decrease in conductivities. In other words, the conductivity stabilizes the G state. However, when the thickness of the insulating films is reduced from 0.12 μm ($l = 3.24\,\mu m$) to 0.05 μm ($l = 3.1\,\mu m$) ((iii) in Fig. 10c), similar behaviour is observed, implying that the insulating film also stabilizes the grid state. If the conductivity is small or the film is thin, the grid state disappears ((iv) in Fig. 10c). These results clearly indicate that the enhanced conductivity of the NLC and the insulating CYTOP film play a crucial role in forming the G state.

To confirm these results, we test typical materials with negative dielectric anisotropy, such as MBBA (Sigma-Aldrich) and an NLC mixture, Phase 5 (Merck). They do not show normal anchoring on the CYTOP surface. For further experimental

evidence, we need to use LC materials that induce homeotropic alignment on CYTOP.

In summary, we report unconventional pattern formation in NLCs by combining doped ions and a perfluoro polymer alignment layer. The creation of a large single domain of square arrays consisting of high-density defects is demonstrated in several ways. The system has huge advantages compared with previous systems because its self-organization offers highly tunable structures that do not require special surface modifications. The structure can be used directly for diffractive microlens arrays, generation of multiple vortex beams using LC mesophases[36,61–65] as foreseen applications. Because our system is self-repairing, it could be applied for sensor applications using director distortion. Polymerizing the structure would provide a soft lithographic template for micro and nanostructures[66]. The stabilization of the grids may offer very interesting possibility to create freestanding films which provide exposed interfaces and are useful, such as for pixelated LC sensor applications in liquid or gaseous environments. Moreover, the system offers a playground for studying the fundamental physics of various fields such as microrheology and colloidal science. Our experimental results may shed light on creating unconventional LC textures using ionic effects. Although the data presented here are obtained under standard laboratory conditions, we believe that the quality can be improved substantially under clean and refined experimental conditions.

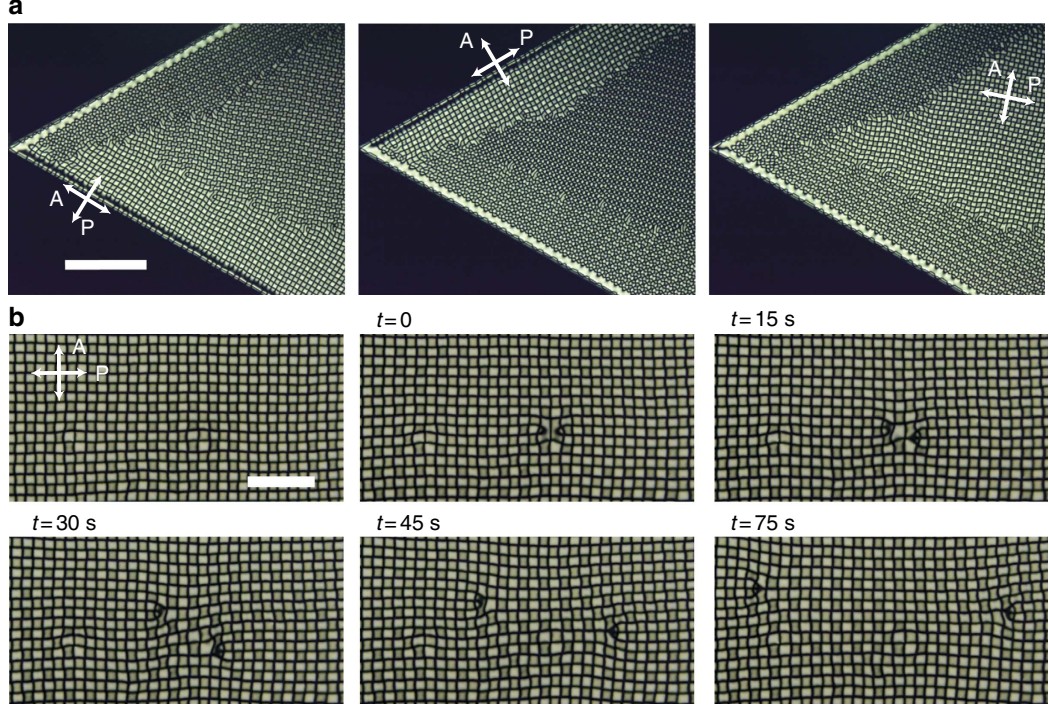

**Figure 9 | Dislocations formed by imperfect arrangements of umbilics. (a)** Three domains of defect arrays with different orientations observed near a sharp corner of the intersected stripe electrode. Each image is taken by rotating polarizers. (**b**) The process of annihilation of a pair of defects and the dynamics of dislocations which repel each other. The frustration is generated by increasing the applied voltage from $V_0 = 17.5$ to $35$ V. The cell thickness is $3.0\,\mu m$ on average and the NLC sample is the 1:1 mixture of CCN-47 and CCN-55. Scale bars, $100\,\mu m$.

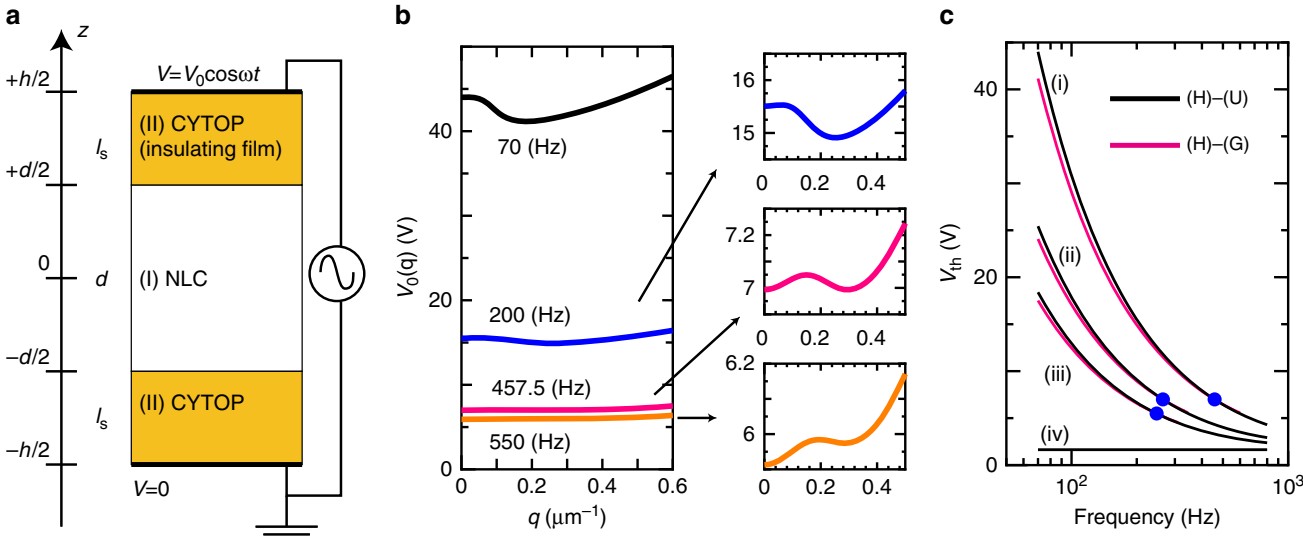

**Figure 10 | Theoretical approach for the grid-like state. (a)** Schematic of our sample cell and the coordinates used for the calculation. $z = \pm d/2$ are the CYTOP–NLC interfaces and $z = \pm h/2$ are the CYTOP-electrode interfaces. The CYTOP thickness $l_s$ is written as $l_s = (h - d)/2$. The electric potential is zero at $z = -h/2$ and $V_0 \cos \omega t$ at $z = h/2$. (**b**) Behaviour of $V_0 (q)$ obtained for four frequencies of 70 Hz (black curve), 200 Hz (blue), 457.5 Hz (magenta) and 550 Hz (orange). The material constants used are: $\varepsilon_{\parallel} = 4$, $\varepsilon_{\perp} = 11$, $\sigma_{\parallel} = 2.6 \times 10^{-6}\,\Omega^{-1}m^{-1}$, $\sigma_{\perp} = 2.1 \times 10^{-6}\,\Omega^{-1}m^{-1}$, $K_1 = 4.5$ pN, $K_3 = 8.5$ pN, $\varepsilon_s = 2$, $\sigma_s = 10^{-15}\,\Omega^{-1}m^{-1}$, $d = 3\,\mu m$ and $h = 3.24\,\mu m$. (**c**) Dependence of $V_{th}$ on frequency $f$ under different conditions: (i) the material constants used are the same with that in **b**, (ii) same with (i) except for $\sigma_{\parallel} = 1.5 \times 10^{-6}\,\Omega^{-1}m^{-1}$, $\sigma_{\perp} = 1.2 \times 10^{-6}\,\Omega^{-1}m^{-1}$, (iii) same with (i) except for $h = 3.1\,\mu m$ and (iv) same with (i) except for $h = d = 3\,\mu m$. Both threshold voltages for the normal Fréedericksz transition (black lines) and the transition of forming the grid pattern (magenta lines) are plotted.

## Methods

**Sample preparation and cells.** We use the NLC compounds CCN-mn (trans, trans-4, 4′-dialkyl-(1α, 1′α-bicyclohexyl-4β-carbonitriles, Nematel GmbH & Co. KG), which possess a negative dielectric anisotropy[47]. Particularly, CCN-37 and the 1:1 mixture of CCN-47 and CCN-55 are used in this work because they show the nematic phase at room temperature. Experiments are carried out at 25 °C unless otherwise indicated. Moreover, 1 wt% of an ionic compound tetrabutyl ammonium benzoate (TBABE, Aldrich) is mixed into NLCs. In preparing the ion-doped samples, the NLC and ions are diluted with chloroform and mixed by ultrasonic agitation for 1 h. Then, the chloroform is evaporated for 24 h.

The sample is filled in cells consisting of ITO coated glass substrates with a proper distance determined by standard interferometry. Monodisperse spherical particles are used as a spacer to maintain the cell thickness. The ITO-coated glass substrate is spin-coated by a thin layer of the amorphous perfluoro polymer (CYTOP, Asahi Glass Co.), which induces the perpendicular orientation of **n** to the glass surface[44]. In the process of our standard spin-coating, we mix the solute CTX-809A (a solution containing 9 wt% CYTOP, Asahi Glass Co.) and the solvent CT-Solv.180 (Asahi Glass Co.) with the weight ratio of 1:2. The spin-coating is made with 800 r.p.m. for 9 s and 3,000 r.p.m. for 15 s. After coating, the substrates are dried at 70 °C for 30 min and baked at 120 °C for longer than 30 min. The ratio of the solute and solvent is changed to obtain various thicknesses of the alignment layers. The thickness of the CYTOP layer is estimated by spectroscopic ellipsometry (SE-2000, SEMILAB Zrt.), whose results are 120 nm for 3 wt% solution and 17 nm for 1 wt% solution. As alternative alignment layers for homeotropic anchoring, a polyimide surface (SE-1211, Nissan chemical) and a surfactant mono layer (DMOAP, N,N-Dimethyl-N-octadecyl-3-aminopropyltrimethoxysilyl chloride, Aldrich) are tested. In the standard use of SE-1211, the stock solution is diluted with the dedicated solvent of equal amount. Spin-coating condition is the same as the CYTOP. In addition, the stock solution of SE-1211 itself is also coated with 1,000 r.p.m. to have a thicker alignment layer although a drastic change is not observed. In this condition, the thickness of SE-1211 amounts to 400 nm. For the latter case, glass substrates are taken into a water solution and 5 vol% DMOAP solution is added to it. After mixing for about 5 min, the excess surfactant is washed with pure water and the substrates are kept at 120 °C for 30 min for drying. The effective electrical conductivity is estimated by a LCR metre (E4980A, Agilent) using an ITO-coated glass cell without having alignment layers. It must be noted that the anisotropy is not accessible. The measured value is of the order of $10^{-6} \, \Omega^{-1} \, m^{-1}$ at 1 kHz, which is used for the theoretical calculation.

To prepare patterned electrodes, ITO-coated glasses are fabricated by a standard photo etching method using a positive photoresist (TFR-2950 PM, Tokyo Ohka Kogyo Co., Ltd.). Finally, an AC voltage $V = V_0 \cos(2\pi f t)$ is applied between the ITO-coated glass substrates along the z-direction in order to reorient the director. For shear application, the upper and lower glass substrates are installed on the motorized stage separately without using spacers.

**Polarization light microscope characterization.** AC voltage from the function generator is amplified and applied to ITO-coated glasses so that the electric field is perpendicular to the substrates. The maximum amplitude of the output voltage $V_0$ is 40 V. Texture observation is usually made by a polarizing microscope (Olympus BX51) under different illumination conditions. If necessary, the temperature controller is used. The micrographic appearance is taken by a DSL camera (Sony ILCE-7R). For our optical manipulation experiments, an Nd-YAG laser (1,064 nm) is irradiated to the sample cell on a motorized stage of an inverted microscope (Olympus IX71). The manipulation is made by moving the motorized stage.

**Fluorescence confocal polarizing microscopy.** A confocal laser scanning microscope Leica TCS sp8 is used for elucidating the director configuration of the G state. The NLC samples are doped with $\sim 0.01$ wt% of a fluorescent dye, 7-diethylamino-3,4-benzophenoxazine-2-one (Nile red, Sigma-Aldrich). The Nile red molecules orient along the director field due to its shape anisotropy. Thin quartz glass plates of thickness 160 μm are used for sample cells. The laser with the wavelength of 552 nm excites the dye molecule and the emission is detected in the spectral region of 610–660 nm. The polarization of the incoming laser beam is adjusted manually with a combination of a quarter wave plate and a linear polarizer inserted in the laser path. Simultaneously, polarizing microscopy images by the ultraviolet transmitted mode are captured during confocal scans of the same sample.

**Theoretical details.** The charge density, $\rho$, satisfies the Poisson equation $\nabla \cdot (\varepsilon \mathbf{E}) = \rho$ and the charge conservation law $\nabla \cdot (\sigma \mathbf{E}) = -\partial \rho / \partial t$, where **E** is the electric field, $\varepsilon$ is the permittivity and $\sigma$ is the conductivity. Here, the ion diffusion is neglected for simplicity. By using the electric potential $\phi$, these equations are rewritten as

$$\nabla \cdot (\varepsilon \nabla \phi) = -\rho, \tag{4}$$

$$\nabla \cdot (\sigma \nabla \phi) = \frac{\partial \rho}{\partial t}. \tag{5}$$

In NLCs, $\varepsilon$ and $\sigma$ are functions of **n**

$$\varepsilon_{\alpha\beta} = \varepsilon_\perp \delta_{\alpha\beta} + \varepsilon_a n_\alpha n_\beta, \tag{6}$$

$$\sigma_{\alpha\beta} = \sigma_\perp \delta_{\alpha\beta} + \sigma_a n_\alpha n_\beta, \tag{7}$$

with $\varepsilon_a = \varepsilon_\| - \varepsilon_\perp$ and $\sigma_a = \sigma_\| - \sigma_\perp$, where $\|$ and $\perp$ denote the components of permittivity (or conductivity) parallel and perpendicular to **n**, respectively. $\varepsilon_s$ and $\sigma_s$ are the permittivity and conductivity of the insulating film, which also imposes a condition of $\sigma_s \ll \sigma_\|, \sigma_\perp$.

On the other hand, director **n** follows the torque balance equation

$$\gamma_1 \mathbf{n} \times \frac{\partial \mathbf{n}}{\partial t} = -\mathbf{n} \times \frac{\delta F}{\delta \mathbf{n}}, \tag{8}$$

where free energy $F$ consists of the Frank elastic energy, $f_d$ and the electric-field contribution, $f_{el}$ as

$$f_d = \frac{K_1}{2} (\nabla \cdot \mathbf{n})^2 + \frac{K_2}{2} (\mathbf{n} \cdot (\nabla \times \mathbf{n}))^2 + \frac{K_3}{2} (\mathbf{n} \times (\nabla \times \mathbf{n}))^2, \tag{9}$$

$$f_{el} = -\frac{1}{2} \varepsilon_\perp E^2 - \frac{1}{2} \varepsilon_a (\mathbf{n} \cdot \mathbf{E})^2, \tag{10}$$

We neglect the contribution of the NLC flow because there is no observable EHC effect, except for in Fig. 8b. $\varepsilon$ and $\sigma$ are time-independent constants in equations (4) and (5) because the relaxation time of the director fluctuations near the threshold voltage is much longer than the period of the AC electric field. Under these conditions, we obtain from equations (4) and (5)

$$\nabla \cdot \left( \tilde{\sigma} \nabla \tilde{\phi} \right) = 0, \tag{11}$$

with

$$\tilde{\sigma} = \sigma + i\omega\varepsilon. \tag{12}$$

Application of Gauss's theorem to equation (11) at the NLC-CYTOP interface gives the boundary conditions of

$$\left( \tilde{\sigma} \nabla \tilde{\phi} \right)_z^{(I)} = \left( \tilde{\sigma} \nabla \tilde{\phi} \right)_z^{(II)} \text{ at } z = \pm d/2, \tag{13}$$

in addition to the continuity of potential of

$$\tilde{\phi}^{(I)} = \tilde{\phi}^{(II)} \text{ at } z = \pm d/2, \tag{14}$$

where superscripts (I) and (II) denote the NLC and CYTOP, respectively, and are used hereafter.

For the homeotropic state, we can easily obtain potential $\tilde{\phi}_0$ from the above equations. The corresponding electric field has only the z component $\tilde{E}_0$ of

$$\tilde{E}_0 = \frac{\tilde{\sigma}_s}{\tilde{\sigma}_\|(h-d) + \tilde{\sigma}_s d} V_0 \text{ in region (I)}, \tag{15}$$

$$\tilde{E}_0 = \frac{\tilde{\sigma}_\|}{\tilde{\sigma}_\|(h-d) + \tilde{\sigma}_s d} V_0 \text{ in region (II)}, \tag{16}$$

where $\tilde{\sigma}_\| = \sigma_\| + i\omega\varepsilon_\|$ and $\tilde{\sigma}_s = \sigma_s + i\omega\varepsilon_s$. Substituting $\mathbf{n} = (\delta n_x, \delta n_y, 1)$ and $\tilde{\phi} = \tilde{\phi}_0 + \delta\tilde{\phi}$ into equations (8)–(14), and linearizing the results with respect to $\delta n_x, \delta n_y$ and $\delta\tilde{\phi}$, we obtain

$$\gamma_1 \frac{\partial \delta n_x}{\partial t} = K_1 \left( \frac{\partial^2 \delta n_x}{\partial x^2} + \frac{\partial^2 \delta n_y}{\partial x \partial y} \right) + K_2 \left( \frac{\partial^2 \delta n_x}{\partial y^2} - \frac{\partial^2 \delta n_y}{\partial x \partial y} \right) + K_3 \frac{\partial^2 \delta n_x}{\partial z^2} \\ - \frac{\varepsilon_a}{2} \text{Re} \left[ |\tilde{E}_0|^2 \delta n_x + \tilde{E}_0^* \frac{\partial \delta\tilde{\phi}}{\partial x} \right] \text{ in (I)}, \tag{17}$$

$$\gamma_1 \frac{\partial \delta n_y}{\partial t} = K_1 \left( \frac{\partial^2 \delta n_x}{\partial x \partial y} + \frac{\partial^2 \delta n_y}{\partial y^2} \right) - K_2 \left( \frac{\partial^2 \delta n_x}{\partial x \partial y} - \frac{\partial^2 \delta n_y}{\partial x \partial y} \right) + K_3 \frac{\partial^2 \delta n_y}{\partial z^2} \\ - \frac{\varepsilon_a}{2} \text{Re} \left[ |\tilde{E}_0|^2 \delta n_y + \tilde{E}_0^* \frac{\partial \delta\tilde{\phi}}{\partial y} \right] \text{ in (II)}, \tag{18}$$

$$\delta n_x = \delta n_y = 0 \text{ at } z = \pm d/2, \tag{19}$$

$$\tilde{\sigma}_\perp \left( \frac{\partial^2 \delta\tilde{\phi}}{\partial x^2} + \frac{\partial^2 \delta\tilde{\phi}}{\partial y^2} \right) + \tilde{\sigma}_\| \frac{\partial^2 \delta\tilde{\phi}}{\partial z^2} - \tilde{\sigma}_a \tilde{E}_0 \left( \frac{\partial \delta n_x}{\partial x} + \frac{\partial \delta n_y}{\partial y} \right) = 0 \text{ in (I)}, \tag{20}$$

$$\Delta \delta\tilde{\phi} = 0 \text{ in (II)}, \tag{21}$$

$$\tilde{\sigma}_\| \frac{\partial \delta\tilde{\phi}^{(I)}}{\partial z} = \tilde{\sigma}_s \frac{\partial \delta\tilde{\phi}^{(II)}}{\partial z}, \delta\tilde{\phi}^{(I)} = \delta\tilde{\phi}^{(II)} \text{ at } z = \pm d/2, \tag{22}$$

$$\delta\tilde{\phi} = 0 \text{ at } z = \pm h/2, \tag{23}$$

where $\tilde{\sigma}_\perp = \sigma_\perp + i\omega\varepsilon_\perp$ and $\tilde{\sigma}_a = \sigma_a + i\omega\varepsilon_a$. The last terms of equations (17) and (18) are replaced by the time average over the period of the applied voltage because the director fluctuations become slow near the threshold.

The translational symmetry in the x-y plane allows us to write the solution for the grid pattern as equations (1)–(3). Substitution of equations (1)–(3) into

equations (17)–(23) yields

$$\gamma_1 \frac{\partial \theta}{\partial t} = -\left(2K_1 q^2 + \frac{\varepsilon_a}{2}|\bar{E}_0|^2\right) + K_3 \frac{\partial^2 \theta}{\partial z^2} - \frac{\varepsilon_a}{2} q \mathrm{Re}\left[\bar{E}_0^* \tilde{\psi}\right] \text{ in (I)}, \quad (24)$$

$$\theta = 0 \text{ at } z = \pm d/2, \quad (25)$$

$$-2q^2 \tilde{\sigma}_\perp \tilde{\psi} + \tilde{\sigma}_\parallel \frac{\partial^2 \tilde{\psi}}{\partial z^2} - 2\tilde{\sigma}_a \bar{E}_0 \theta = 0 \text{ in (I)} \quad (26)$$

$$\frac{\partial^2 \tilde{\psi}}{\partial z^2} = 0 \text{ in (II)} \quad (27)$$

Equations (17) and (18) are reduced to the same equation (24), and the boundary conditions for $\tilde{\psi}$ are obtained by replacing $\delta\tilde{\phi}$ with $\tilde{\psi}$ in equations (22) and (23). At threshold voltage $V_{th}$, the relaxation time diverges so that the time derivative term on the left-hand side of equation (24) vanishes. Then, equation (24) with $\frac{\partial \theta}{\partial t} = 0$, together with the other equations (25)–(27) gives $V_{th}$. The above equations (24)–(27) cannot be solved analytically for general cases $h \neq d$, except for $h = d$. Then, we briefly explain how to calculate $V_{th}$ numerically. Expressing $\tilde{\psi} = \psi' + i\psi''$, we have three equations for three quantities $\theta, \psi'$ and $\psi''$ from equations (24)–(27). Substitution of $\theta = \theta_0 e^{\lambda z}$, $\psi' = \psi_0' e^{\lambda z}$, and $\psi'' = \psi_0'' e^{\lambda z}$ gives a system of homogeneous linear equations for $\mathbf{u} = (\theta_0, \psi_0', \psi_0'')$, which can only be solved if the determinant vanishes. Because there are two derivatives, $\frac{\partial^2 \theta}{\partial z^2}$ and $\frac{\partial^2 \tilde{\psi}}{\partial z^2}$, in equations (24)–(27), we have a cubic equation with respect to $\lambda^2$. Furthermore, from the symmetry of the above equations (24)–(27), the solutions can be classified into odd and even functions of $z$, and the latter gives the minimum threshold voltage. Thus, the solution can be expressed as $\mathbf{u} = c_1 \mathbf{u}_1 \cosh\lambda_1 z + c_2 \mathbf{u}_2 \cosh\lambda_2 z + c_3 \mathbf{u}_3 \cosh\lambda_3 z$, where $\lambda_i^2$ is the eigenvalue and $\mathbf{u}_i$ is the corresponding eigenvector. In the insulating film ($d/2 < z < h/2$), whereas the potential is obtained as $\tilde{\psi} = (c_4 + ic_5)\sinh\sqrt{2}q(z - h/2)$ after imposing the boundary condition $\tilde{\psi}(h/2) = 0$. Substitution of these equations into the other boundary conditions yields a system of homogeneous linear equations for $c_i$, the determinant of which is a function of $V_0$ and $q$, and vanishes at the threshold voltage for a given $q$.

**Data availability.** The authors declare that the data supporting the findings of this study are available within the article and its Supplementary Information files.

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

## Acknowledgements

This work was supported by JSPS KAKENHI Grant Number JP25103006 and by the Foundation for the Promotion of Ion Engineering.

## Author contributions

Y.S., V.S.R.J., C.T., N.S., S.S., K.V.L. and F.A. performed experiments. H.O. performed theoretical analysis. V.S.R.J., F.A. and H.O. designed the project. Y.S., V.S.R.J., K.V.L., F.A. and H.O. wrote the manuscript. All authors have seen and approved the final manuscript.

## Additional information

**Competing financial interests:** The authors declare no competing financial interests.

