## [Peer review file · Nature Communications]

Reviewer #1 (Remarks to the Author):

The manuscript by Sasaki et al is an exciting piece of research reporting on a totally new phenomenon and an approach to generate defect arrays in liquid crystals. Namely, the authors demonstrate that by using a special alignment layer and by doping a liquid crystal of a negative dielectric anisotropy with ionic additive, they can generate a number of defect patterns, including very regularly spaced umbilics and stripes. The findings are striking and will definitely attract a lot of interest to this paper. However, the presentation should be improved.

1. The authors described the field-induced states as dielectrically/conductivity controlled structures. It might well be the case, but the readers would probably be skeptical and there must be some place in the experimental part that clearly states (with added evidence) why the director deformations cannot be related to (a) electrohydrodynamics, (b) flexoelectricity; (c) surface polarization.
2. The authors are advised to check the formula for the threshold voltage on page 4; I am not sure where the coefficient 2 comes from.
3. I am not sure the umbilics are of a pure splay type (page 6); the lambda plate cannot discriminate between purely radial and somewhat spiralling director; the statement about "confirmation" of splay should be mitigated.
4. FCPM technique should be accompanied by references that describe the essence of this microscopy approach.
5. The text should define I_s .
6. The authors should use SI units
7. The formula for V_{th} on page 10 needs to be checked; the presence of "d" in the second term makes it dimensionally wrong; this misprint also suggests that the authors need to explain how the formula has been derived.
8. The thickness of the alignment layer represents an important parameter in the discussion of the phenomenon and its origin. It is estimated as being rather thick, on the order of one micrometer. Yet the authors make no attempt to determine it experimentally, instead citing other works that have no impact on their actual experiments. The thickness measurement does not appear to pose any problem; the simplest approach might be to use a profilometer or to scratch the layer and to explore it under an AFM.
9. In general, the text focuses a bit too much on the auxiliary features such as dislocation dynamics at the expense of a more detailed description of the phenomenon of defect patterns appearance and its analysis; the auxiliary sections can be shortened (or presented in the future publications) and the freed space can be used to address more fully points such as pp 1, 7 and 8 above.

Reviewer #2 (Remarks to the Author):

This work proposes the electrical generation of large-scale generation of nicely ordered arrays of umbilical liquid crystal defects in nematic liquid crystals planar films provided with perpendicular surface orientational boundary conditions. The specificity of the proposed approach is that it does not rely on a pre-patterned "field" crystal (for instance alignment, electric field, or optical field) imposing a desired topology to the liquid crystal that eventually leads to the generation of a defect. The mechanism at play is identified to be the presence of ion-doping of the liquid crystal material with the use of insulating surface boundary conditions. The generation of such defect arrays is associated with thorough experimental investigations of possible ways to reconfigure the arrays, by electrical, optical and hydrodynamic means. The interesting presence and dynamics of "crystallographic" defects in arrays of liquid crystal defects is also discussed. The role of the geometry of the domain contour is particularly fascinating. A model is also proposed to describe quantitatively the dependence of orientational effects as a function of the voltage and frequency of the applied electric field. As such, this work represents a substantial progress in the generation of reconfigurable arrays of topological textures in liquid crystal materials that deserves publication.

It might be interesting to:

- mention/discuss possible realization of large arrays of umbilics with one free surface. Indeed, such situation could have interesting application potential to develop pixelated liquid crystal sensors of molecular compounds in liquid or gaseous environments.

- quantify such "positional noise" and discuss it in the framework of 2D crystal effective temperature.

Regarding the application potential mentioned in the manuscript in the context of multiple-vortex beam generation, it should be noted that has been addressed in a few previous works, using various liquid crystal mesophases. Therefore, it would make sense to position the present work accordingly:

- Cholesteric liquid crystals: in work [J. Opt. 15, 044021 (2013)], with arbitrary arrays of light-induced defects.

- Smectic liquid crystals: in work [Opt. Express 22, 4699 (2014)], with large ordered arrays of spontaneously occurring defects.

- Nematic liquid crystals: in work [Appl. Phys. Lett. 105, 121108 (2014)], with one-dimensional arrays of electrically induced defects.

Minor note: "perfectly" sounds too much enthusiastic in the abstract since images shows long-range imperfections even in absence of defects in the array itself.

point-by-point answers to comments

For Reviewer #1:

The manuscript by Sasaki et al is an exciting piece of research reporting on a totally new phenomenon and an approach to generate defect arrays in liquid crystals. Namely, the authors demonstrate that by using a special alignment layer and by doping a liquid crystal of a negative dielectric anisotropy with ionic additive, they can generate a number of defect patterns, including very regularly spaced umbilics and stripes. The findings are striking and will definitely attract a lot of interest to this paper. However, the presentation should be improved.

[Response] Thank you very much for reviewing our manuscript. We highly appreciate the reviewer's recognition of the importance of our work and the positive comment. We have answered the comments by the reviewer point by point below. Besides, according to the reviewer's suggestion, we have modified the manuscript to improve the presentation as clearly as possible. We would be very grateful if our modifications and replies could be suitable.

1. The authors described the field-induced states as dielectrically/conductivity controlled structures. It might well be the case, but the readers would probably be skeptical and there must be some place in the experimental part that clearly states (with added evidence) why the director deformations cannot be related to (a) electrohydrodynamics, (b) flexoelectricity; (c) surface polarization.

[Response] We thank the reviewer for this insightful comment for improving the quality of the manuscript. As described in the manuscript, we believe that the phenomenon is obtained by the dielectric property rather than the electrohydrodynamics, flexoelectricity, and surface polarization. The main discussion is based on the fact that the threshold voltages in high frequency region show a good consensus among sample cells irrespective of the ionic impurity and the alignment layers. In order to more explicitly explain that the mechanism originates from the Fréedericksz transition, we have modified the overall description in the section "**Behavior of threshold voltages**" with newly-conducted experimental data. **(Revision [2])** In Figure 3, we have added a threshold curve for 1 wt% CYTOP solution in order to make the results more convincing. Actually, we have some more data plots filling in the blanks of the graph, but we think that this is already enough. We have also mentioned the essential difference from other suspicious effects in the micrographic texture and in the behavior of the threshold voltage. Some references have been added to cite earlier reports (Monkade, M. *et al. Europhys. Lett.* **2**, 299 (1986), Lavrentovich, O. D. *et al. Phys. Rev. A* **45**, R6969 (1992), Blinov, L. M. *J. Phys. Colloq.* **40**, C3–247 (1979)) which show the director change at surfaces. We hope you find that our explanation is clear and acceptable.

2. *The authors are advised to check the formula for the threshold voltage on page 4; I am not sure where the coefficient 2 comes from.*

[Response] In our work, the applied voltage is defined as $V = V_0 \cos 2\pi ft$. The Fréedericksz transition upon AC field application is usually expressed using the root-mean-square value $V_0/\sqrt{2}$, so that the factor 2 appears.

3. *I am not sure the umbilics are of a pure splay type (page 6); the lambda plate cannot discriminate between purely radial and somewhat spiralling director; the statement about "confirmation" of splay should be mitigated.*

[Response] We agree with your comment. The use of the lambda plate provides only qualitative information, and does not allow us to judge the detailed structure. Since the text was a bit misleading, we have modified the statement. **(Revision [3])**

4. *FCPM technique should be accompanied by references that describe the essence of this microscopy approach.*

[Response] Thank you very much for the suggestion. In accordance with the reviewer's suggestion, we have added two additional references (Smalyukh, I. I. *et al.* Chem. Phys. Lett. 336, 88–96 (2001), Smalyukh, I. I. *et al.* Phys. Rev. Lett. 96, 177801 (2006)) which describe the essence of FCPM. **(Revision [4])**

5. *The text should define l_s .*

[Response] We have defined l_s in the text as well as in the caption of Figure 3. **(Revision [2][6])**

6. *The authors should use SI units*

[Response] We have changed $\Omega \cdot \text{cm}$ to $\Omega \cdot \text{m}$ in the revised manuscript. **(Revision [2][9])**

7. *The formula for V_{th} on page 10 needs to be checked; the presence of "d" in the second term makes it dimensionally wrong; this misprint also suggests that the authors need to explain how the formula has been derived.*

[Response] Thank you very much for pointing out our mistake. We regret this confusion. Because of this mistake, we have changed the first paragraph in the **Discussion** part. Also the text has been revised by including newly obtained spectroscopic ellipsometry data. We believe that this

modification is appropriate and makes the text more understandable. (Revision [9])

8. *The thickness of the alignment layer represents an important parameter in the discussion of the phenomenon and its origin. It is estimated as being rather thick, on the order of one micrometer. Yet the authors make no attempt to determine it experimentally, instead citing other works that have no impact on their actual experiments. The thickness measurement does not appear to pose any problem; the simplest approach might be to use a profilometer or to scratch the layer and to explore it under an AFM.*

[Response] We thank the reviewer for the kind suggestion. We have evaluated the thickness of the alignment layer by the newly-conducted spectroscopic ellipsometry experiments, which revealed that the thickness is thinner than those we assumed in the previous manuscript. Importantly, this fact strongly indicates that the CYTOP has a considerable effect on electrical insulation of ions compared to the polyimide surface. So we are really grateful to the reviewer for the proper advice to improve the results. Based on the newly obtained data, the theoretical parts have been modified. We hope the revised manuscript is now more convincing. (Revision [2][9][10][11][13])

9. *In general, the text focuses a bit too much on the auxiliary features such as dislocation dynamics at the expense of a more detailed description of the phenomenon of defect patterns appearance and its analysis; the auxiliary sections can be shortened (or presented in the future publications) and the freed space can be used to address more fully points such as pp 1, 7 and 8 above.*

[Response] We thank the reviewer for the suggestion. We have tried to shorten and simplify the auxiliary sections. Finally, the section of “**Annihilation of defects through destabilization**” has been removed because this manuscript is focused mainly on the structure of newly-found regular defect arrays and their stabilization. The results on the destabilization can be reported elsewhere in near future with more intensive information. The name of auxiliary sections has been changed to “**tunable lattice spacing**”. Moreover, the Supplementary Movie 9 on defect-annihilation process has been deleted. (Revision [7][8] [15])

point-by-point answers to comments

For Reviewer #2:

This work proposes the electrical generation of large-scale generation of nicely ordered arrays of umbilical liquid crystal defects in nematic liquid crystals planar films provided with perpendicular surface orientational boundary conditions. The specificity of the proposed approach is that it does not rely on a pre-patterned "field" crystal (for instance alignment, electric field, or optical field) imposing a desired topology to the liquid crystal that eventually leads to the generation of a defect. The mechanism at play is identified to be the presence of ion-doping of the liquid crystal material with the use of insulating surface boundary conditions. The generation of such defect arrays is associated with thorough experimental investigations of possible ways to reconfigure the arrays, by electrical, optical and hydrodynamic means. The interesting presence and dynamics of "crystallographic" defects in arrays of liquid crystal defects is also discussed. The role of the geometry of the domain contour is particularly fascinating. A model is also proposed to describe quantitatively the dependence of orientational effects as a function of the voltage and frequency of the applied electric field. As such, this work represents a substantial progress in the generation of reconfigurable arrays of topological textures in liquid crystal materials that deserves publication.

[Response] Thank you very much for reviewing our manuscript and for providing useful suggestions and encouraging comments. Based on the reviewer's comment, we revised the manuscript to enhance the quality of manuscript as much as possible. We hope the current version of the manuscript is acceptable.

It might be interesting to:

- mention/discuss possible realization of large arrays of umbilics with one free surface. Indeed, such situation could have interesting application potential to develop pixelated liquid crystal sensors of molecular compounds in liquid or gaseous environments.*
- quantify such "positional noise" and discuss it in the framework of 2D crystal effective temperature.*

[Response] (1) We highly appreciate the reviewer's suggestion with a very interesting example toward applied research. We realize that the reviewer's idea is extremely interesting and important for the future research. Since the application point of view is also necessary for fascinatingly presenting such a new physical phenomenon, we would like to incorporate the reviewer's idea of pixelated liquid crystals sensors. **(Revision [12])** We would like to mention that such single- or double-side free surface structures could be realized by soft-polymerization of the patterned structure. Then, the film with the pinned grids can be further investigated in various environments.

(2) Regarding the quantification of the "positional noise", we are very sorry for not providing the

quantitative data in this current manuscript and it seems difficult to answer the raised question here. We have tried to estimate the deviation of the position of defects from the ideal lattice structure. But we found that the behavior of dislocations in a local area is complicated, although the general tendency shows that the total number of defects decreases over time. Probably, this is because even the ideal state contains some long-ranged deformations as you pointed out, which arise from some imperfections in the sample cell. So, in such a case, it seems that we need to quantify both the density difference and the long-ranged deformation. However, in this moment, we could not find an appropriate answer. This could be investigated in detail in future. We would be very grateful if you could understand the situation.

Regarding the application potential mentioned in the manuscript in the context of multiple-vortex beam generation, it should be noted that has been addressed in a few previous works, using various liquid crystal mesophases. Therefore, it would make sense to position the present work accordingly:

- *Cholesteric liquid crystals: in work [J. Opt. 15, 044021 (2013)], with arbitrary arrays of light-induced defects.*
- *Smectic liquid crystals: in work [Opt. Express 22, 4699 (2014)], with large ordered arrays of spontaneously occurring defects.*
- *Nematic liquid crystals: in work [Appl. Phys. Lett. 105, 121108 (2014)], with one-dimensional arrays of electrically induced defects.*

[Response] Thank you very much for the references. We fully agree with your suggestion. These papers have been added in the revised manuscript. **(Revision [12])**

Minor note: "perfectly" sounds too much enthusiastic in the abstract since images shows long-range imperfections even in absence of defects in the array itself.

[Response] We agree with your suggestion. “**perfectly ordered**” is a bit exaggerated as the reviewer pointed out. So we have changed the words to “**regularly ordered**” at two places. **(Revision [1][5])**

Reviewer #1 (Remarks to the Author):

The revised manuscript addresses my original comments and can be published.

point-by-point answers to comments

For Reviewer #1:

The revised manuscript addresses my original comments and can be published.

[Response] Thank you very much for reviewing our manuscript. We highly appreciate the reviewer's comment for encouraging publication.